# Immuno-detection by sequencing enables large-scale high-dimensional phenotyping in cells

Jessie A.G. van Buggenum [ID] [1], Jan P. Gerlach[1], Sabine E.J. Tanis[1], Mark Hogeweg[1,2], Pascal W.T.C. Jansen[3], Jesse Middelwijk[4], Ruud van der Steen[4], Michiel Vermeulen[3], Hendrik G. Stunnenberg[3], Cornelis A. Albers[1,2] & Klaas W. Mulder[1]

Cell-based small molecule screening is an effective strategy leading to new medicines. Scientists in the pharmaceutical industry as well as in academia have made tremendous progress in developing both large-scale and smaller-scale screening assays. However, an accessible and universal technology for measuring large numbers of molecular and cellular phenotypes in many samples in parallel is not available. Here we present the immuno-detection by sequencing (ID-seq) technology that combines antibody-based protein detection and DNA-sequencing via DNA-tagged antibodies. We use ID-seq to simultaneously measure 70 (phospho-)proteins in primary human epidermal stem cells to screen the effects of ~300 kinase inhibitor probes to characterise the role of 225 kinases. The results show an association between decreased mTOR signalling and increased differentiation and uncover 13 kinases potentially regulating epidermal renewal through distinct mechanisms. Taken together, our work establishes ID-seq as a flexible solution for large-scale high-dimensional phenotyping in fixed cell populations.

[1] Department of Molecular Developmental Biology, Radboud Institute for Molecular Life Sciences, Radboud University, PO Box 9101, 6500 HB Nijmegen, The Netherlands. [2] Department of Human Genetics, Donders Institute for Brain, Cognition and Behaviour, Radboud University Medical Center, PO Box 9101, 6500 HB Nijmegen, The Netherlands. [3] The Oncode Institute, Department of Molecular Biology, Radboud Institute for Molecular Life Sciences, Radboud University, PO Box 9101, 6500 HB Nijmegen, The Netherlands. [4] Biolegio BV, PO Box 916500 AB Nijmegen, The Netherlands. Correspondence and requests for materials should be addressed to K.W.M. (email: k.mulder@science.ru.nl)

Quantification of protein levels and phosphorylation events is central to investigating the cellular response to perturbations such as drug treatment or genetic defects. This is particularly important for cell-based phenotypic screens to discover novel drug leads in the pharmaceutical industry. However, the complexity of biological and disease processes is not easily captured by changes in individual markers. Currently, a major limitation is the trade-off between the number of samples and the number of (phospho-)proteins that can be measured in a single experiment. For instance, immunohistochemistry (IHC)[1] and immunofluorescence (IF)[2] allow high-throughput protein measurements using fluorescently labelled antibodies. However, these methods are limited in the number of (phospho-)proteins that can be measured simultaneously in each sample due to spectral overlap of the fluorescent reporter dyes. One commercial solution, Luminex®, has circumvented this limitation by using colour-barcoded antibody-loaded beads and allows multiplexing of some 50 proteins per sample[3–5]. However, this approach requires cell lysis and does currently not include phospho-specific signalling detection. Several alternative approaches based on antibody–DNA conjugates have been developed in recent years[6, 7]. For instance, Ullal et al. used the Nanostring system to quantify 88 antibody–single-strand DNA (ssDNA) oligo conjugates in fine needle aspirates[6]. Although powerful, this strategy is not well suited for high-throughput applications. Furthermore, the commercial Proseek® strategy entails a proximity extension assay using pairs of ssDNA oligo coupled antibodies in combination with quantitative PCR as a read-out[7]. This assay is generally performed on cell lysates and currently there are no assays for phospho-proteins available to study signalling activity. In addition, several other recently described antibody–DNA conjugate-based methods that use high-throughput sequencing as a read-out detect only a few extracellular epitopes or at low sample throughput[8–12], limiting their scope. Here we present immuno-detection by sequencing (ID-seq) as a streamlined universal technology for measuring large numbers of molecular phenotypes, for many samples in parallel. We show that high-throughput sequencing of antibody-coupled DNA barcodes allows accurate and reproducible quantification of 84 (phospho-)proteins in hundreds of samples simultaneously. We apply ID-seq in conjunction with the published kinase inhibitor set (PKIS) to start investigating the role of >200 kinases in primary human epidermal stem cell renewal and differentiation. This demonstrates a downregulation of mammalian target-of-rapamycin (mTOR) signalling during differentiation and uncoveres 13 kinases potentially regulating epidermal renewal through distinct mechanisms.

## Results

**Precise and sensitive (phospho-)protein detection**. We designed the ID-seq technology to simultaneously measure many proteins and post-translational modifications in high-throughput (Fig. 1a). At the basis of ID-seq lie antibodies that are labelled with a double-stranded DNA (dsDNA) tag[13] containing a 10-nucleotide antibody-dedicated barcode and a 15-nucleotide unique molecular identifier (UMI, Supplementary Fig. 1, Supplementary note 1). Each antibody signal is now digitised and non-overlapping, allowing many antibodies to be combined and measured simultaneously. Following immunostaining and washing, DNA barcodes are released from the antibodies through reduction of a chemically cleavable linker[13] and a sample-specific barcode is added through PCR. Finally, samples are pooled to prepare an indexed sequencing library (Fig. 1a, Supplementary Fig. 1 and Supplementary note 1). This triple barcoding strategy facilitates straightforward incorporation of hundreds (and potentially

thousands) of samples per experiment and achieves count-based quantification (Supplementary Fig. 2 and Supplementary note 2) with a dynamic range of four orders of magnitude (Supplementary Fig. 3). Furthermore, analyses of 17 antibody–DNA conjugates using singleplex and multiplexed measurements show high correspondence ($R = 0.98 \pm 0.046$), demonstrating that multiplexing does not interfere with antibody detection (Fig. 1b). Moreover, the ID-seq library preparation procedure is reproducible ($R = 0.98$, Fig. 1c) and precise, as determined using nine distinct DNA tag sequences per antibody, serving as technical replicates ($R > 0.99$, Supplementary Fig. 4). Finally, cell-dilution series and small interfering RNA (siRNA)-mediated silencing of selected proteins showed an epitope abundance-dependent decrease of antibody-barcode counts, confirming the specificity of the ID-seq signals (Supplementary Fig. 5 a, b). Collectively, these experiments show that the ID-seq technology allows precise, sensitive and specific multiplexed protein quantification through sequencing antibody-coupled DNA tags.

**Constructing a 70 antibody–DNA conjugate ID-seq panel**. To fully exploit the multiplexing capacity of ID-seq, we obtained 111 antibodies targeting intracellular and extracellular epitopes. We aimed to generate an antibody panel that will allow us to determine the state of the cell in a broad manner and where possible should be reactive towards both human and mouse epitopes. Therefore, the initial selection of antibodies covered a wide range of cellular processes, including cell cycle, DNA damage, epidermal self-renewal and differentiation, as well as the intracellular signalling status for the epidermal growth factor (EGF), G-protein-coupled receptors, calcium signalling, tumour necrosis factor-α (TNFα), transforming growth factor-β (TGFβ), Notch, WNT and BMP pathways, and can potentially be applied to a variety of cellular systems. All of the selected antibodies were validated for specificity in IF, IHC and/or fluorescence-activated cell sorting applications by the vendor (see Supplementary Data 1 for details and links to datasheets). As these applications include cell fixation, we reasoned that this selection would increase the chance of identifying antibodies that are suitable for ID-seq. From these 111 antibodies, 84 showed sound signals in in-cell-western/IF and/or immuno-PCR experiments using antibody dilutions and/or IgG control antibodies (Supplementary Figs. 6 and 7). To increase our confidence in this set of antibodies, we performed a series of experiments using IF, immuno-PCR and/or ID-seq as a read-out to verify that the tested antibodies show the expected signal dynamics in response to specific perturbations. These perturbations included: induction of differentiation; stimulation with EGF or bone morphogenetic proteins (BMPs); induction of DNA damage signalling with mitomycin C or hydroxyurea, as well as inhibition of EGF and BMP signalling with the small molecule inhibitors AG1478 and DMH1, respectively (Supplementary Fig. 8). Also, signals of a subset of phospho-specific antibodies were decreased upon phosphatase treatment of fixed cell populations (Supplementary Fig. 9 and Supplementary Data 1). Taken together, 64 out of the 84 antibodies exhibited the expected protein or phospho-protein dynamics in our primary skin stem cells in these experiments, whereas the rest was stable, indicating their utility in ID-seq. The 84 antibodies displayed ~75-fold signal over no-cell background, a measure of technical noise (Supplementary Fig. 10). Moreover, we found that the variability of the signals from a subpanel of 69 antibody–DNA conjugates was below 20% among 14 biological replicates (coefficient of variation < 0.2, Fig. 1d), demonstrating the precision and reproducibility of highly multiplexed ID-seq measurements. These experiments enabled us to construct a panel of ~70 antibody–DNA conjugates to evaluate protein levels and

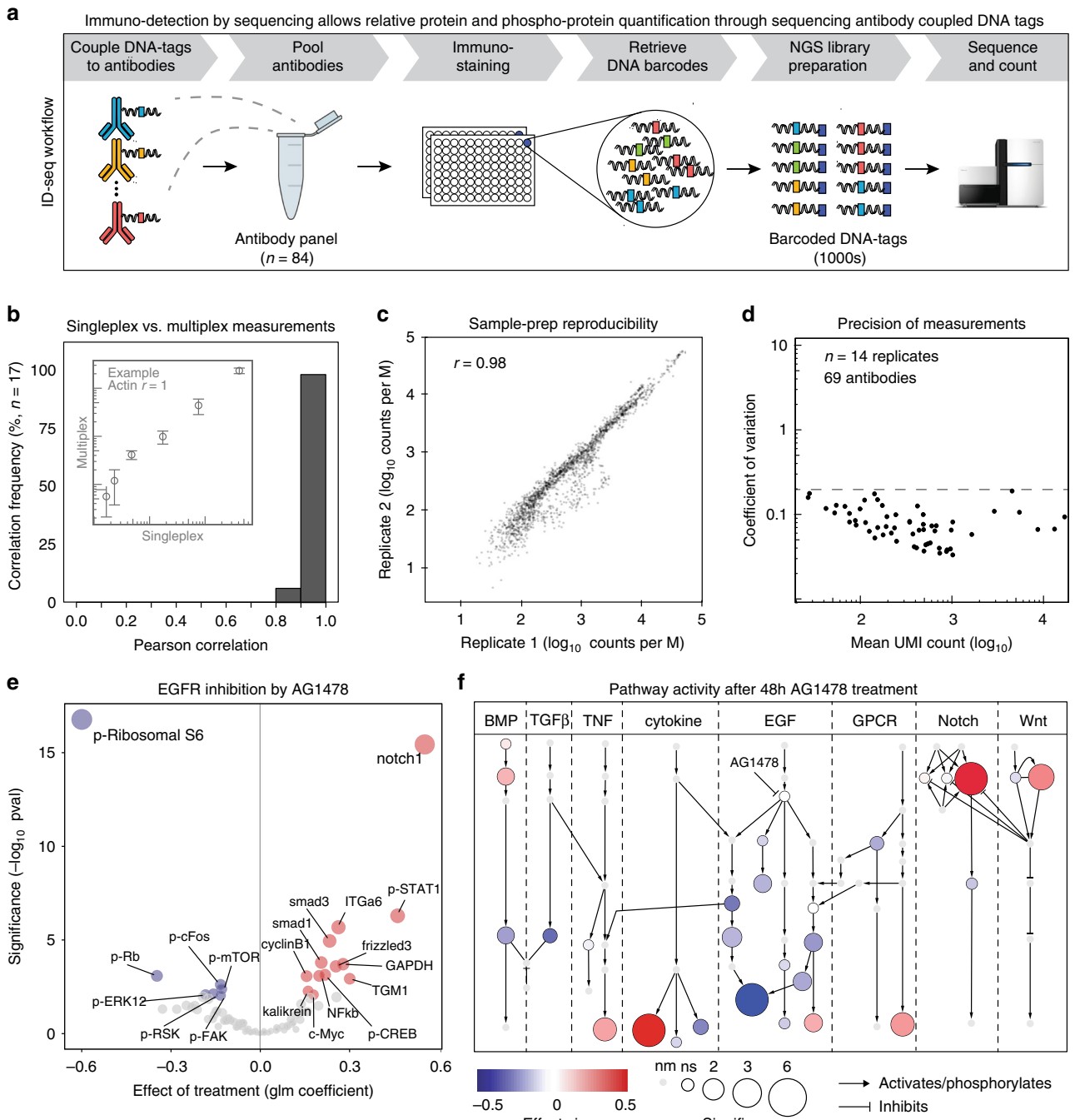

**Fig. 1** Immuno-detection by sequencing (ID-seq) technology development. **a** Concept of the ID-seq technology. First, pool DNA-tagged antibodies. Second, perform multiplexed immunostaining on fixed cell populations, and release DNA tags. Third, barcode the released DNA tags through a two-step PCR protocol. Finally, sequence the barcoded DNA tags via next-generation sequencing (NGS) and count barcodes. **b** Signals from singleplex epitope detection (via an immuno-PCR measurement) were compared with multiplexed epitope detection using ID-seq. The histogram summarises correlations between 17 immuno-PCR and corresponding ID-seq measurements. Insert panel illustrates an example from the actin antibody showing signal mean and s.d. ($n = 4$). Underlying data for all 17 antibodies can be found in Supplementary Fig. 4. **c** Scatterplot indicates the high reproducibility of PCR-based ID-seq library preparation ($r =$ Pearson correlation). Libraries from the same released material were prepared on separate occasions and analysed in different sequencing runs. **d** The scatterplot shows the counts (mean) and coefficient of variation from 69 antibody–DNA conjugates ($n = 14$ biological replicates). Dashed line indicates 20% variation. **e** Volcano plot shows the effect (estimate) and significance ($-\log_{10}$ pval) of AG1478 treatment ($n = 6$), based on the model analysis of ID-seq counts (Supplementary Note 3). Significance ($-\log_{10}$ pval) determines node size. Red nodes show significantly increased (ANOVA, $p < 0.01$) and blue nodes show significantly decreased (ANOVA, $p < 0.01$) (phosphor-)protein levels. **f** Pathway overview of ID-seq measurements after 48 h of AG1478 treatment. Colour indicates the effect size and node size represents the significance of effect ($-\log_{10}$ pval). Light grey nodes without border indicate not measured (nm) proteins (see Supplementary Supplementary Fig. 12 for (phospho-)protein identities)

intracellular signalling, covering a broad range of biological processes, including cell cycle, apoptosis, DNA damage and cell-type-specific epidermal self-renewal and differentiation. Also, the panel covers intracellular signalling pathways EGF, G-protein-coupled receptors, calcium signalling, TNFα, TGFβ, NOTCH, BMP and WNT pathways (Supplementary Data 1). Of note, the nature of the selected and validated antibodies should make this panel broadly applicable to many other human (and mouse)-derived cell systems, and other antibodies can be added when required.

**Measuring (phospho-)protein levels in human skin stem cells.** Primary human epidermal stem cells (keratinocytes) depend on active EGF receptor (EGFR) signalling for self-renewal in vitro and vivo[14]. We inhibited this pathway using the potent and selective inhibitor AG1478 at a concentration suitable for cell-based assays (10 µM, 48 h) to determine whether ID-seq recapitulates keratinocyte biology. In these experiments, we would expect to at least observe dynamic changes in downstream EGFR signalling pathway activity, as well as in the expression of differentiation-associated proteins. To analyse the effects of AG1478 treatment on each antibody signal, we developed a generalised linear mixed (glm) model that takes into account the negative binomial distribution of ID-seq count data and

incorporates potential sources of variation (e.g., replicates, batches and sequencing depth). This model derives the effect ('estimate') of treatment on each antibody, followed by a like-lihood ratio test to determine the significance of the effect (Supplementary note 3). We identified 13 increased and 7 decreased (phospho-)proteins upon AG1478 treatment ($p < 0.01$, analysis of variance, Fig. 1e). Upregulation of the known differentiation markers transglutaminase 1 (TGM1) and NOTCH1 confirmed successful differentiation. Although the induction of late differentiation marker TGM1 was relatively modest at this early stage of differentiation (48 h of EGFR inhibition), quantitative proteomics, IF and quantitative reverse transcription PCR (RT-qPCR) measurements showed comparable increases of TGM1 levels (Supplementary Fig. 11). This indicates that modest, yet biologically informative, effects can be identified using the ID-seq technology.

Next, we projected the estimates and significance levels of our ID-seq results onto a literature-derived signalling network (Fig. 1f, see Supplementary Fig. 12a for node identities). As expected, EGFR pathway activity was downregulated upon AG1478 treatment. We also identified effects on the activity of several other pathways, including the BMP and Notch cascades, which are known players in epidermal biology[15-18]. RT-qPCR analysis revealed that these effects arose from changes in mRNA

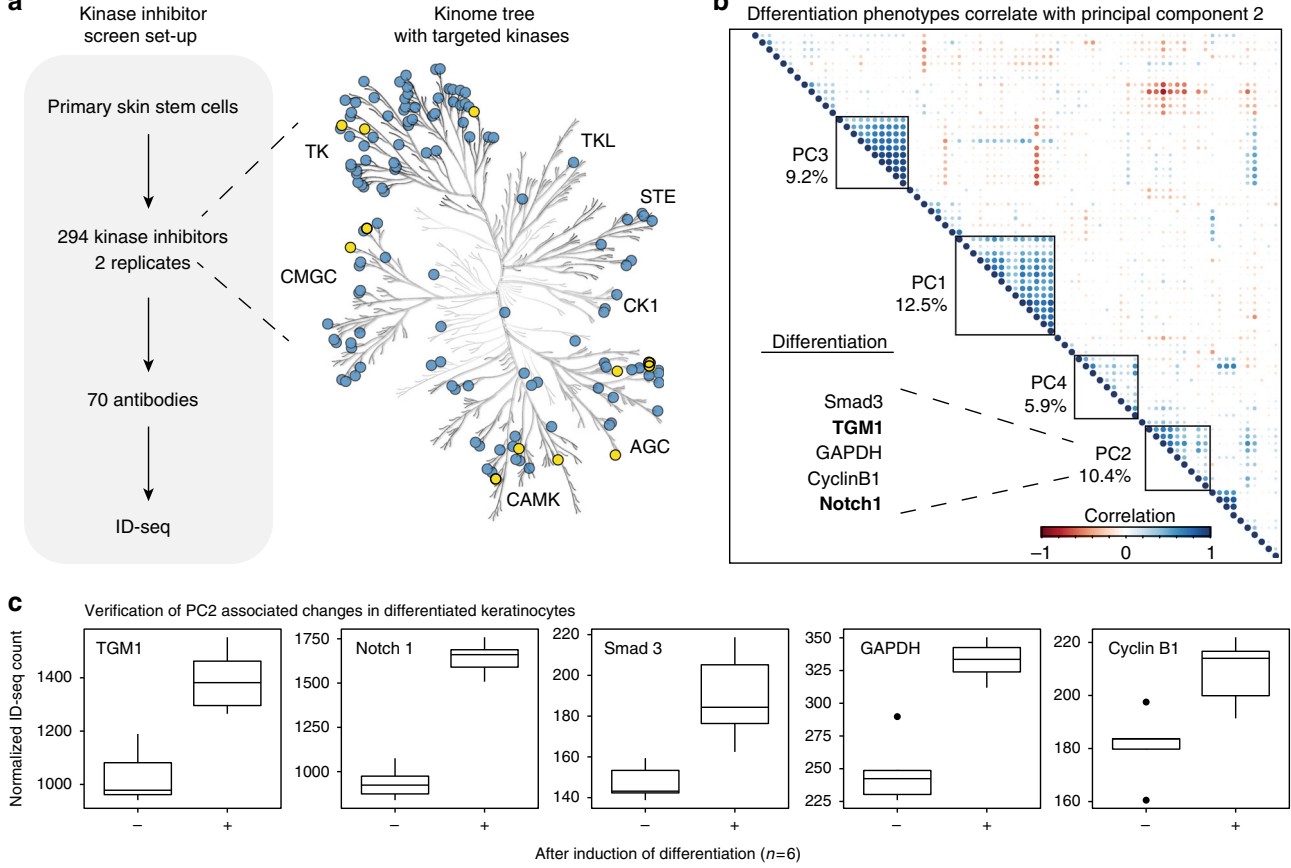

**Fig. 2** ID-seq screen of PKIS identifies probes inducing skin stem cell differentiation. **a** Schematic overview of the protein kinase inhibitor set (PKIS) screen set-up. Kinase tree shows kinases targeted by the inhibitory probes (blue) and significantly enriched kinases (yellow, Fig. 4a). The probes target all major kinase families (TKL, STE, CK, AGC, CAMK, CMGC and TK). **b** To combine PKIS probe effects on multiple ID-seq molecular phenotypes to one measure, we performed principal component analysis on the PKIS data set using the signed $\log_{10}$ $p$-values of the ID-seq analysis. Then, we clustered all molecular phenotypes and the top five PCs to identify the PC summarising differentiation of the skin stem cells (in bold molecular phenotypes TGM1 and NOTCH1). **c** ID-seq measurement of differentiation marker TGM1, Notch 1, GAPDH, Cyclin B1 and SMAD3 upon inhibition of EGFR shows differentiation-induced phenotypic changes. (Boxplots with centre line indicating median, bounds of boxes showing upper and lower quartile, and whiskers illustrating 1.5 × interquartile range, $n = 6$)

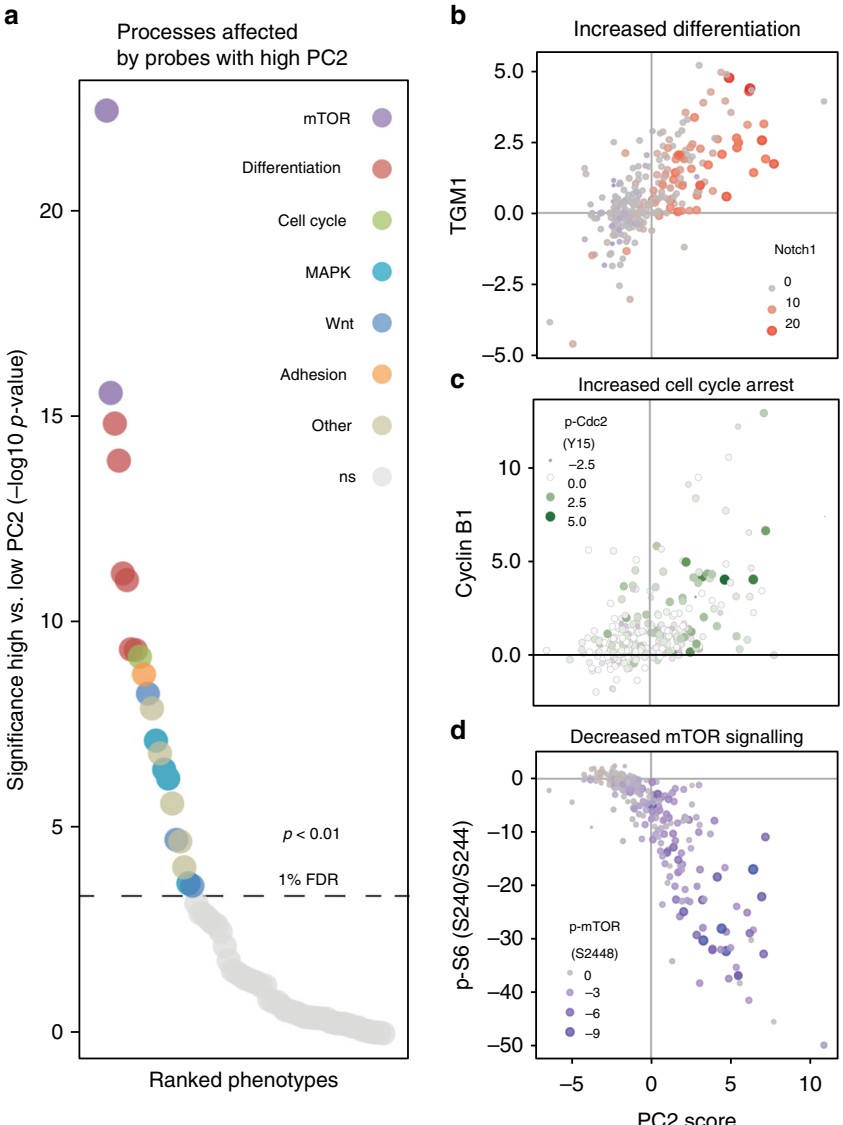

**Fig. 3** Probes with high PC2 affect differentiation, cell-cycle arrest and mTOR signalling activity. **a** Summary of significantly ($p < 0.01$, FDR 1%, $t$-test, Supplementary Fig. 17) affected molecular phenotypes from probes with high PC2 (top 10%) compared to low PC2 (bottom 10%). **b** Scatterplot illustrating probes with high PC2 score ($x$-axis) have increased TGM1 and NOTCH1 levels measured by ID-seq. **c** Scatterplot illustrating probes with high PC2 score ($x$-axis) have increased cell-cycle arrest marker (Cyclin B1 and p-cdc2) levels measured by ID-seq. **d** Scatterplot showing the probes with high PC2 score ($x$-axis) strongly have decreased phospho-S6 and phospho-mTOR levels illustrating decreased mTOR signalling activity measured by ID-seq

expression of BMP ligands and NOTCH receptors (Supplementary Fig. 12b). We confirmed activation of the BMP and Notch pathways by RT-qPCR analysis of their classical downstream target genes *ID2* and *HES2*, respectively (Supplementary Fig. 12b). These results demonstrate the potential of ID-seq and our glm model to distinguish different treatment conditions by quantifying changes in (phospho-)protein dynamics.

**Kinase inhibitors induce skin stem cell differentiation**. As we are able to interrogate the complex biological process of keratinocyte differentiation underlying EGFR, we decided to expand our search. Extracellular signals involved in epidermal renewal and differentiation are widely studied and include EGF, TGFβ, BMP, Notch ligands and Wnts[15, 19–23]. However, the contributions of different intracellular effector kinases on renewal and differentiation are not well documented. To start addressing this issue, we applied ID-seq to human epidermal keratinocytes

treated with the PKIS, an open-source chemical probe library[24–26] containing ~300 small molecules targeting 225 kinases across all major kinase families in the human proteome (Fig. 2a). To determine the effects of kinase inhibition at the molecular level, we performed ID-seq with a panel of 70 antibodies on cells seeded in 384-well plates and treated with 294 PKIS compounds for 24 h. Replicate screens were highly correlated ($R = 0.98$) and had low UMI duplicate rates (1.2%), indicating high data quality (Supplementary Fig. 13a-e). We annotated the effect and its significance for each kinase inhibitor probe on each of the measured molecular phenotypes (as measured by our antibody conjugates) and used the results of this analysis to interrogate the effect of kinase inhibition on skin stem cell biology.

A key advantage of the high multiplexing capacity of ID-seq is its potential to simultaneously measure multiple antibodies reflecting a given biological process. We anticipate this to result in a more reliable and comprehensive measurement of the affected processes compared to quantification of a single marker.

We exploited the multiplexed nature of the ID-seq data by combining the individual phenotypic ID-seq measurements into principal components (PCs) through PC analysis (PCA). The PCA essentially aggregates the molecular phenotypes that jointly explain independent fractions of variation in the data into a single score, which in turn represents the effect of the inhibitory probes on the skin stem cells. To determine the underlying processes associated with each PC per probe, we correlated and clustered the measured antibodies with the top four PCs explaining 38% of total variation in the data set (Fig. 2b, Supplementary Fig. 14). As expected from a screen using kinase inhibitor probes a considerable fraction of this variation is associated with effects on signalling pathway activity phenotypes, as represented by PC1 (Fig. 2b). In line with this, our ID-seq antibody panel contained several up- and downstream components of the pathways involving some of the kinases targeted by groups of compounds in the PKIS library. We confirmed that these groups of inhibitors indeed affect their expected read-outs, where upstream regulators showed increased signals and downstream targets showed decreased signals in our screen (Supplementary Fig. 15a-c). The second largest PC identified in our analysis, PC2, strongly correlates with proteins that are significantly upregulated upon differentiation, including the known marker proteins TGM1 and NOTCH1 (Fig. 2b and Supplementary Fig. 14[15, 27]). Interestingly, Cyclin B1, GAPDH and SMAD3 were also included in this cluster. We confirmed that changes of these molecular phenotypes genuinely reflect keratinocyte differentiation, by forcing the cells to differentiate using the EGFR inhibitor AG1478 for 48 h and subjecting these samples to ID-seq ($n = 6$). Indeed, protein levels of TGM1, NOTCH1, SMAD3, Cyclin B1 and GAPDH are upregulated upon differentiation, corroborating the results of our screen (Fig. 2c). To validate these screen results further, we selected 18 probes that showed high PC2 scores from the PKIS library for colony formation experiments, the gold standard in vitro assay for epidermal stem cell proliferation (Supplementary Fig. 16a, b[28],) combined with IF measurement of differentiation marker TGM1. Automated image analysis was used to quantify colony number, colony size (and size distribution), as well as the level of the differentiation marker TGM1 level per colony for each probe ($n = 3$ replicates). Fifteen out of the eighteen tested probes showed a significant effect on at least one of the measured colony phenotypes (Supplementary Fig. 16a, b). This indicates that high-PC2 probes indeed affect epidermal cell colony-forming capacity and authenticates the PC2 score as a bona fide reflection of differentiation.

**Dynamic molecular processes in differentiated skin cells**. We gathered that the top and bottom 10% of PC2-ranked probes are likely to distinguish the differentiating (high-PC2) and non-differentiated (low-PC2) epidermal cell states. To determine which molecular processes are different between these two cell states, we identified the molecular phenotypes that display a significant increase or decrease ($p < 0.01$, 1% false discovery rate (FDR), $t$-test) between cell populations with high PC2 vs. low PC2. This revealed differential levels of the Wnt pathway (measured by Fzd3 and phosphorylated-LRP6), MAPK signalling (phospho-p38, phospho-SRC, phospho-cFOS and phospho-RSK), integrin-mediated adhesion (phospho-FAK) and the mTOR pathway (phospho-mTOR and phospho-S6) (Fig. 3a and Supplementary Fig. 17). Plotting the PC2 scores vs. the ID-seq measurements of differentiation markers revealed that high-PC2 probes indeed display increased differentiation marker expression (Fig. 3b), and have increased cell-cycle arrest markers (Fig. 3c). Indeed, keratinocyte differentiation is associated with a G2/M cell-cycle arrest in vivo and vitro[29–31]. Strikingly, all high-PC2

probes have a strong downregulation of the mTOR pathway activity, suggesting a role for this pathway in epidermal biology (Fig. 3d). To confirm the suggestion of decreased mTOR signalling in differentiating keratinocytes, we used an independent approach to induce differentiation and performed RT-qPCR and IF/in-cell western analysis of differentiating keratinocytes (Supplementary Fig. 18). We induced differentiation by growing cells at increasing cell density to induce a range of differentiation levels. Subsequent IF measurements of TGM1, as well as RT-qPCR analysis of *TGM1* and Periplakin (*PPL*) confirmed the induction of differentiation of these samples (Supplementary Fig. 18a, b). Consistent with the observation of inverse correlation between mTOR signalling levels and differentiation markers in the PKIS screen results, we observed an inverse correlation between ribosomal S6 protein phosphorylation and cell density, confirming that decreased mTOR signalling is associated with differentiation (Supplementary Fig. 18a). Moreover, we found that mRNA expression levels of the mTOR co-factor *RAPTOR*, but not of *mTOR* itself, decreased concordantly with the drop in S6 phosphorylation levels, suggesting that decreased mTOR signalling activity is potentially caused by decreased *RAPTOR* gene expression (Supplementary Fig. 18b). Of note, even though decreased RAPTOR mRNA levels were associated with differentiation, we found that siRNA-mediated silencing of RAPTOR on its own was not sufficient to cause cells to differentiate, as assessed by RT-qPCR analysis of several key differentiation markers (Supplementary Fig. 18c). Together, the ID-seq PKIS screen uncovered relevant molecular phenotypes associated with keratinocyte differentiation, including a decrease in mTOR signalling activity.

**Identification of kinases involved in skin stem cell renewal**. We reasoned that the inhibited kinases that strongly associated with PC2 are likely involved in epidermal stem cell renewal, as their inhibition leads to increased differentiation and cell-cycle arrest. To determine which kinases are inhibited by probes with high PC2 scores, we made use of available data on the biochemical selectivity and potency of the PKIS compounds towards 225 individual kinases[26]. We applied outlier statistics to assign a set of inhibitory probes to each of these 225 kinases ($p < 0.01$, Supplementary Data 2). As indicated above, these probe sets show expected effects on targeted or downstream signalling molecules (Supplementary Fig. 15). Subsequent gene set enrichment analysis (GSEA) identified 13 probe sets that were enriched ($p < 0.01$, 1% FDR) in PC2 (i.e., probes inducing cell differentiation and/or cell-cycle arrest) and of which the corresponding kinase is expressed in keratinocytes (Fig. 4, Supplementary Figs. 19, 20). Our analysis returned the EGFR as the top hit, reflecting its recognised importance in epidermal stem cell renewal in vitro and in vivo. The probes that inhibit the other kinases are distinct from those inhibiting the EGFR, indicating that the identification of these 12 kinases did not result from cross-reactivity of the probes towards the EGFR (Supplementary Fig. 21a, b). Important to note is the potential of each probe to inhibit more than one kinase (Supplementary Fig. 21a), as the PKIS probe library was designed as a platform for lead discovery and to provide chemical scaffolds for further medicinal chemistry[24–26]. Additional to the EGFR, the list of identified kinases included PRKD3 and FYN, two intracellular kinases shown to impact epidermal biology[32–36]. Interestingly, p70S6K is the downstream effector of the mTOR/RAPTOR complex that phosphorylates the ribosomal S6 protein. The fact our GSEA analysis identified this kinase matches our finding that mTOR signalling is decreased upon epidermal differentiation (Fig. 3c and Supplementary Fig. 18a-c). Moreover, immuno-histochemical staining of human skin sections showed that the

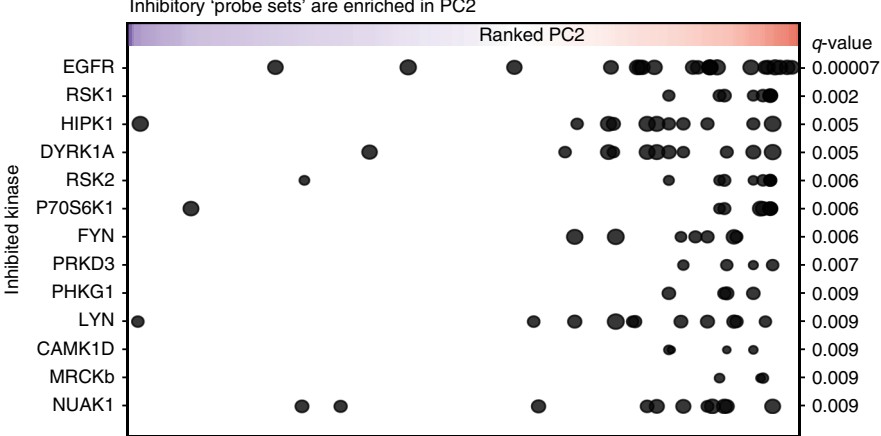

**Fig. 4** GSE analysis identifies kinases targeted by probes leading to epidermal differentiation. Summary plot of GSE analysis of probes in PC2 shows expressed inhibited kinases ordered according to the significance of enrichment ($p$-values (FDR) < 0.01, Supplementary Fig. 22). Probes (points in the graph) are ranked according to PC2 ($x$-axis), and point size shows % of inhibition for the indicated kinase by the probe

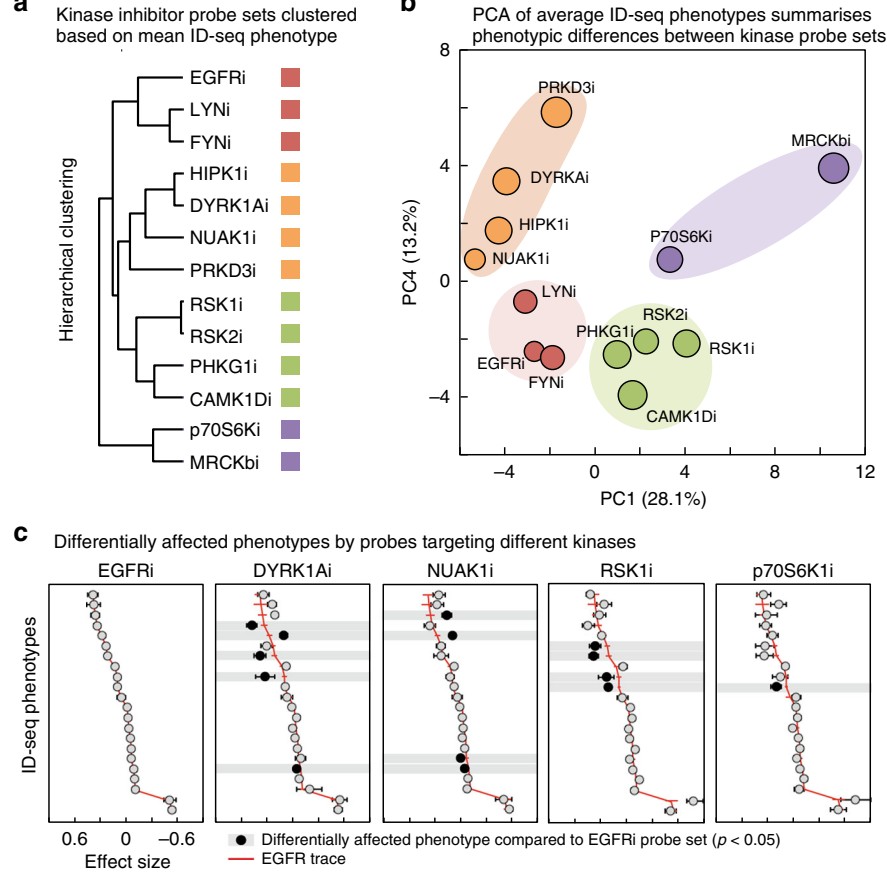

**Fig. 5** Enriched kinase inhibitor probe sets show differentially affected molecular phenotypes. **a** $K$-means clustering of mean probe effect on molecular phenotypes per kinase probe set. **b** Principal component analysis on mean probe effects shows in PC1 and PC4 distinct kinase probe-set clusters. Colours based on $K$-means clustering of the data (see **a**). **c** The mean and standard error of probe effect on molecular phenotypes, per probe set inhibiting kinases EGFR, DYRK1A, NUAK1, RSK1/2/3/4 and p70-Ribosomal S6 kinase. Compared to EGFR the other four kinases have a comparable phenotypic profile with several changes different effects on molecular phenotypes (black nodes, $p < 0.05$, $t$-test)

expression of the EGFR, RSK1 and PHKG1 is restricted to cells residing in the epidermal stem cell niche, whereas NUAK1 is expressed throughout the epidermis (Supplementary Fig. 22), consistent with our findings that inhibition of these kinases affects epidermal stem cell biology. Taken together, ID-seq identified both known and previously unrecognised potential

kinase effectors of epidermal renewal across four major kinase families (Fig. 2a, yellow nodes).

We investigated whether the information contained in the ID-seq data set may be used to explore the underlying molecular mechanism of the kinase inhibition. For each kinase set, we calculated the mean effect on each of the measured molecular

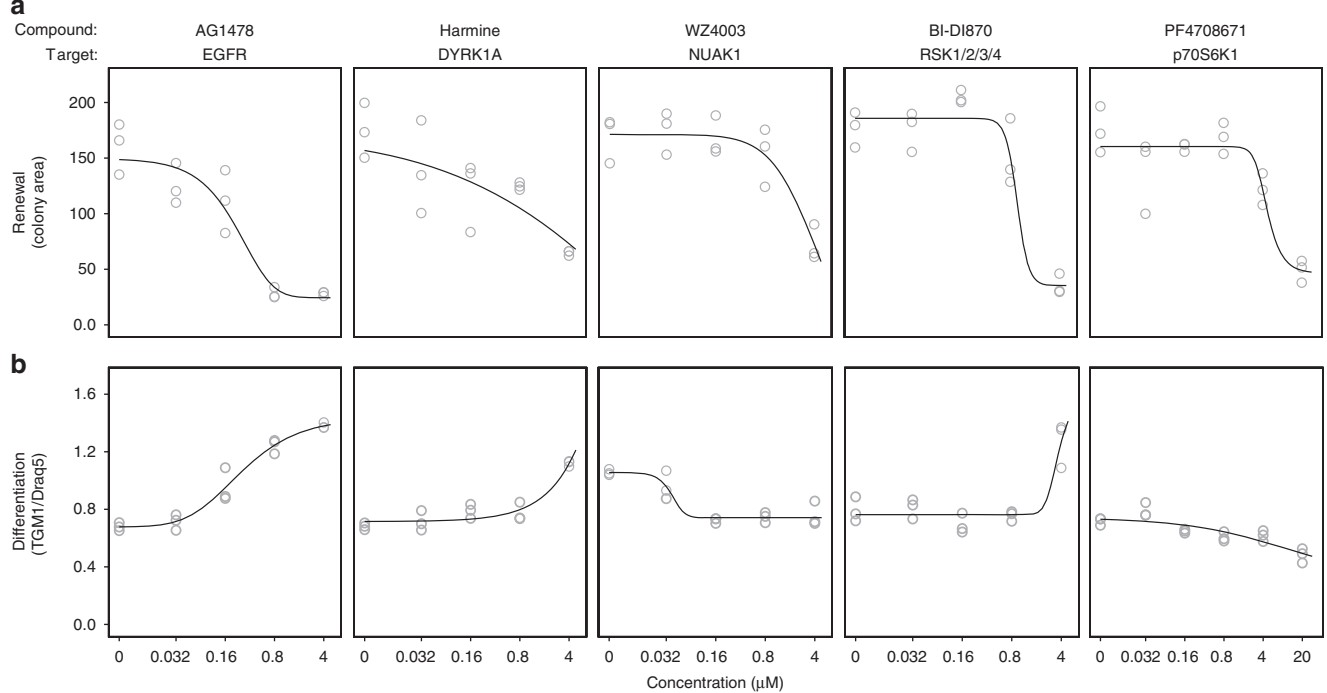

**Fig. 6** Renewal assay confirms crucial role for DYRK1A, NUAK1, RSK and p70S6 kinases in skin stem cell renewal while the inhibitors have distinct effect on late differentiation marker TGM1. **a** Colony area of keratinocytes (mean area of all colonies per replicate) after 9 days of growth in the presence of specific kinase inhibitors AG1478, Harmine, WZ4003, BI-DI870 and PF4708671 targeting EGFR, DYRK1A, NUAK1, RSK1/2/3/4 and p70-Ribosomal S6 kinase, respectively. Line shows modelled dose–response curves based on three biological replicates per concentration. **b** Levels of late differentiation marker (mean of all colonies per replicate), determined by immunofluorescent staining of TGM1 in CFA cell populations (corrected for cell number using Draq5 staining). Modelled dose–response curves ($n = 3$) show increase in TGM1 levels for three out of five inhibitors

phenotypes. These kinase set-level molecular profiles were used for both hierarchical clustering and PCA, separating these 13 enriched kinases into four distinct subgroups (Fig. 5a, b). The EGFR and its immediate downstream kinases, LYN and FYN, form a tight cluster, indicating that this grouping reflects molecular mechanistic relationships. In turn, this predicts that the kinases in the other subgroups may function through mechanisms that are different and potentially independent from the EGFR. If this was indeed the case, we would expect inhibition of these kinases to result in distinctive effects on (subsets of) the interrogated molecular phenotypes. We compared the inhibitory effects (average model-derived estimate ± SEM) on the 20 molecular phenotypes distinguishing differentiated and renewing epidermal cells (as defined in Fig. 2c) for exemplars of these four subgroups of kinases. Ranking these read-outs based on the effect of EGFR inhibition and plotting the data of the other kinases in the same order showed that their overall trends were conserved, reflecting that these probe sets indeed affect epidermal cell differentiation (Fig. 5c). Importantly, most of the kinases showed statistically significant deviations ($p < 0.05$, $t$-test) in discrete subsets of molecular phenotypes compared to the EGFR, indicating that they potentially function through distinct mechanisms to regulate epidermal renewal (Fig. 5c). Together, this analysis shows that ID-seq allows highly multiplexed screening of hundreds of chemical probes, identifying kinases involved in epidermal renewal and at the same time provides information on the underlying molecular mechanism to categorise the identified effector kinases.

Finally, as a first step towards verification of the importance of the kinases representing these four subgroups in epidermal self-renewal, we obtained inhibitors of the EGFR, RSK1-4, p70S6K, NUAK1 and DYRK1A, independent from the PKIS library. These chemical inhibitors were chosen as they were described to display

selectivity for their intended targets, although it is difficult to fully exclude any contribution from (lower affinity) inhibition of unintended targets kinases. We examined the effects of these inhibitors on epidermal stem cell function using in vitro colony formation assays (Fig. 6a, b and Supplementary Fig. 23). For this, stem cells were first allowed to adhere to the culture plate containing a layer of feeder cells. Twenty-four hours after seeding, the cultures were exposed to the kinase inhibitors in a broad range of concentrations for the remainder of the culture period. Subsequent automated imaging-based quantification of the resulting colonies revealed that inhibition of the EGFR, RSK1-4 and DYRK1A decreased the self-renewal capacity of the epidermal stem cells (as determined by colony size) and stimulated the expression of the late differentiation marker TGM1. In contrast, p70S6K and NUAK1 inhibition resulted in decreased renewal but did not increase TGM1 expression. These results support a potential role for these kinases in epidermal renewal, although further work will be required to fully characterise and understand their contributions to this process.

## Discussion

Cell-based phenotypic screens are frequently used in academia and the pharmaceutical industry to identify leads for drug development[37, 38]. However, obtaining insight into the molecular mechanism of action of the selected compound can be time-consuming and expensive[38–40]. We developed the ID-seq technology as an approach to facilitate high-throughput highly multiplexed molecular phenotyping. We showed that ID-seq could be applied to large numbers of samples for precise and sensitive protein measurements in fixed cell populations. The dynamic range of our counting strategy by sequencing is seemingly broader than enzyme-linked immunosorbent assays and

comparable to the other high-throughput screening assays such as the AlphaScreen and the Luminex, respectively[41, 42]. These assays measure proteins in solution, such as body fluids and cell extracts. In addition, the AlphaScreen technology requires pairs of antibodies for each target, whereas ID-seq does not require cell lysis, works on fixed cells in multi-well plates and uses one antibody per target. Therefore, we consider ID-seq as a novel technology that is complementary to existing commercial approaches.

We applied ID-seq to primary human epidermal keratinocytes in conjunction with the PKIS chemical probe library and identified several kinases that are important for epidermal stem cell function. The depth of measured molecular phenotypes in our screen resulted in the identification of the association between decreased mTOR signalling pathway activity and epidermal differentiation. This is in line with recent findings that the mTOR pathway plays an important role in human keratinocyte at the switch from proliferation to differentiation[43]. Additionally, our findings serve as a valuable resource to identify potential drugs that could lead to severe skin toxicity since many targeted therapies in clinical trials are directed against kinases[44, 45]. Thus, ID-seq allows high-throughput molecular screening of kinase inhibitors and leads to meaningful insight in skin biology.

The straightforward ID-seq workflow was designed to be compatible with automation for applications in industry. This scaling potential should enable in-depth analysis of mechanisms of action for 100s, potentially 1000s, of compounds in a single experiment. Beyond the kinase-based screening application presented here, the ID-seq technology can in principle be applied to any cell system, any perturbation and with any validated high-quality antibody, making it a flexible solution for large-scale high-dimensional phenotyping.

## Methods

**Cell culture and transfections**. Keratinocytes (pooled foreskin strain KNP, Lonza) were expanded as described[31] supplemented with Rock inhibitor (Y-27632, 10 μM). After expansion of the keratinocytes on feeders, the cells were grown for 1–3 days on keratinocyte serum-free medium (KSFM) with supplements (bovine pituitary extract (30 μg ml$^{-1}$) and EGF (0.2 ng ml$^{-1}$, Gibco) in a 96- or 384-well plate at ~10.000 cells/well. Before 48 h AG1478 treatment (10 μM), the cells were cultured for 48 h in a 96-well plate. Before EGF stimulation, keratinocytes were grown for 3 days on KSFM with and 1 day without supplements. After starvation, cells were stimulated with EGF (100 ng ml$^{-1}$) for 5 min. For the PKIS screen, 10 000 cells were seeded in a 384-well plate and grown for 24 h in KSFM with supplements, followed by 24-h treatment with PKIS compounds (10 μM) or dimethylsulphoxide (DMSO). siRNA nucleofections were performed with the Amaxa 96-well shuttle system (Lonza). Keratinocytes were grown in KSFM to ~70% confluency, harvested and resuspended in cell line buffer SF. In all, $2 \times 10^5$ cells were used for each 20 μl transfection (programme FF-113) with 1–2 μM siRNA duplexes. Please note that is equivalent to 5–10 nM siRNA in conventional liposome-based transfections. Transfected cells were incubated at ambient temperature for 5–10 min post transfection and subsequently resuspended in pre-warmed KSFM. Silencer Select siRNAs were used throughout this study (Ambion/Applied Biosystems).

**Conjugation of antibodies to dsDNA**. Antibodies and dsDNA were functionalised and conjugated as described[13]. See Supplementary Data 1 for a list of antibodies. In short, antibodies were functionalised with NHS-s-s-PEG4-tetrazine in a ratio of 1:10 in 50 mM borate-buffered saline (BBS), pH 8.4 (150 mM NaCl). Then, N3-dsDNA was produced and functionalized with DBCO-PEG12-TCO (Jena Bioscience). See Supplementary Data 3 for a list of oligo sequences. Finally, purified functionalised antibodies were conjugated to purified functionalised DNA by 4-h incubation at room temperature in borate-buffered saline, pH 8.4, in a ratio of 4:1 respectively. The reaction was quenched with an excess of 3,6-diphenyl tetrazine. After pooling, the conjugates were incubated with ProtA/G beads in BBS overnight. After thorough washes with phosphate-buffered saline (PBS), the conjugates were eluted from the beads with 0.1 M citrate, pH 2.3, and immediately neutralised with Tris-HCl, pH 8.8. Subsequently, a buffer exchange into PBS (pH 7.4) was performed using two Zeba-spin desalting columns. The size of the eluted DNA was checked on an agarose gel, confirming that all unconjugated DNA was removed using this purification approach.

**Antibody characterisation**. Detailed information on the antibodies is summarised in Supplementary Data 1. In brief, we selected antibodies suitable for IHC or IF validated by manufacturer. The Developmental Studies Hybridoma Bank (DSHB) antibodies were produced and purified as described[13]. We choose antibodies to study a wide variety of biological processes as mentioned in the Results section. We performed antibody-dilution series with all antibodies on our primary human keratincoytes to show antibody-dependent signal via IF. Moreover, we coupled antibodies to DNA barcodes as described in our conjugation and immuno-PCR protocol[13]. These antibodies show antibody concentration-dependent signal via immuno-PCR, indicating successful conjugation, release and detection of DNA tags. Finally, we performed several modulation experiments to show specific dynamics in our keratinocytes measured via the antibodies in IF, immuno-PCR (Supplementary Data 1, Supplementary Figs. 6, 7 and 9) or ID-seq (Supplementary Fig. 5 and Supplementary Fig. 8). Supplementary Fig. 5b shows protein levels (measured via ID-seq) and mRNA levels measured via qPCR. qPCR analysis was performed according to standard protocol (iQTM SYBR Green Supermix, CFX 96 machine). qPCR primers are show in Supplementary Table 1.

**Immunostainings and release of DNA tags**. Keratinocytes were fixed with 4% paraformaldehyde in PBS for 15 min at room temperature (RT), washed three times with PBS and stored at 4 °C (up to 3–4 days before further use). Then, cells were permeabilised and blocked for 30 min using 0.5× protein-free blocking buffer (Thermo Fisher) in PBS with 0.1% Triton and 200 ng ml$^{-1}$ single-strand Salmon Sperm DNA (sssDNA). Blocking the cells and wells with sssDNA is crucial to suppress background binding of the antibody–DNA conjugates[13]. Then, cells were incubated with conjugated antibodies in the same buffer at 0.1 μg ml$^{-1}$ antibody, at 4 °C overnight. After immunostaining with conjugates, the cells were thoroughly washed with PBS (3× short, 3× 15 min and 3× short). Then, release buffer was freshly prepared (10 mM dithiothreitol (DTT) in borate-buffered saline, pH 8.4). Cells were incubated with 20–50 μl of release buffer depending on the plate type and well size and incubated for 90 min at RT, with careful mixing (on a vortex) every 30 min. Released DNA barcodes were collected and stored at −20 °C.

**Sample barcoding and sequencing library preparation**. To barcode the released DNA tags from each cell population (see Supplementary Note 1 for sequence design), a 25 μl PCR was performed per sample containing 8–15 μl sample with released DNA tags, 0.2 mM dNTPs, 1 μl PFU polymerase, 1× PFU buffer (20 mM Tris-HCl (pH 8.8), 2 mM MgSO$_4$, 10 mM KCl, 10 mM (NH$_4$)$_2$SO$_4$, 0.1% triton and 0.1 mg ml$^{-1}$ bovine serum albumin), spike-in DNA barcodes, forward primer (AATGATACGGCGACCACCG, Biolegio) and a well-specific reverse primer (Supplementary Data 3, Supplementary Fig. 1b). In a 96-well PCR machine (T100 Thermal Cycler, Biorad) the following programme was used: (1) 3 min at 95 °C; (2) 30 s at 95 °C; (3) 30 s at 60 or 54 °C; (4) 30 s at 72 °C; (5) repeat 2–4 nine times; (6) 5 min at 72 °C; and (7) ∞12 °C. Then, all well-specific labelled DNA barcodes from one plate were pooled to one sample. This sample was purified using a PCR purification column (Qiagen) according to the manufacturer's protocol. Samples were eluted with 30 μl nuclease-free water. To remove any residual primers, samples were treated with Exonuclease I in 1× PFU buffer for 30 min at 37 °C. After inactivation for 20 min at 80 °C, another 25 μl PCR reaction was prepared with 15–17 μl sample, 0.2 mM dNTPs, ×1 μl PFU polymerase, 1 × PFU buffer, forward primer (AATGATACGGCGACCACCG, Biolegio) and a sample-specific reverse primer with Illumina index barcode and adapter sequence (Supplementary Data 3, Supplementary Fig. 1d). The same programme as PCR reaction (I) was used, and reactions were purified over a PCR purification column (Qiagen). All PCR reactions were then incubated for 45 min with 1 μl Exonuclease I to remove residual primers. The PKIS screen samples were further size-selected using size selection columns (Zymo, according to the manufacturer's protocol) for fragments > 150 bp. Finally, all samples were purified over PCR purification mini-elute column (Qiagen) and eluted in 10 μl elution buffer. Final sequencing samples were run on a 2% agarose gel (0.5× TBE) with 10× SYBR Green I (Life Technologies) and scanned on a Typhoon Trio+ machine (GE Healthcare), or analysed with the 2100 Bioanalyzer (Agilent) to confirm the size of the DNA fragments (expected size around 185 bp).

**ID-seq data analysis**. Sequence data from the NextSeq500 (Illumina) were demultiplexed using bcl2fastq software (Illumina). The quality of the sequencing data was evaluated using a FastQC tool (version 0.11.4 and 0.11.5, Babraham Bioinformatics). Then, all reads were processed using our dedicated R-package (IDSeq, Supplementary Note 2). In short, the sequencing reads were split using a common anchor sequence identifying the position of the UMI sequence, barcode 1 (antibody-specific) and barcode 2 (well-specific) sequence. After removing all duplicate reads, the number of UMI sequences were counted per barcode 1 and 2. Finally, barcode 1 and barcode 2 sequences were matched to the corresponding antibody and well information.

Using R-package DESeq2[46], we calculated normalisation factors (estimated size factor) to account for differences in sequencing depth per sample. Using lme4, we analysed the effect of a specific condition using a linear mixed effect model (Supplementary Note 3).

For each antibody in the PKIS screen the effect and significance of each treatment were determined as described in Supplementary Note 3. Then, the 'signed p-value' was derived from the sign of the model estimate (positive/negative) and the p-value. This signed p-value was used as input for the PCAs (Fig. 2b and Supplementary Fig. 15). To calculate effects of probe sets per molecular phenotype, the mean model estimate was calculated. These means were used for subsequent PCA analysis and 'molecular phenotype profiles' described in Fig. 5.

**Immuno-PCR experiments.** The immuno-PCR experiments were performed as described previously in our paper on antibody–DNA conjugates[13]. In short, each antibody was conjugated to dsDNA, and used in an immunostaining as described. DNA was released using 10 mM DTT in BBS, pH 8.4, and measured by quantitative PCR using iQTM SYBR Green Supermix on CFX 96 machine. The $2^{-Ct}$ values were used to calculate the mean signal and standard deviation from four biological replicates. The Pearson correlation between these immuno-PCR and the multiplexed ID-seq signal was calculated using the mean.

**Proteomics.** Cells were harvested, washed, snap-frozen and stored at −80 °C until lysis and mass spectrometry (MS) analysis. Induction of differentiation was validated by qPCR (data not shown) before lysis. Cells were lysed using lysis buffer (4% SDS, 100 mM Tris-HCl (pH 7.6) and 100 mM DTT) and by boiling for 3 min at 95 °C. DNA was sheared using sonication, 5 cycles: 30 s ON and 30 s OFF (high). The samples were centrifuged for 5 min at $16\,000 \times g$, 4 °C and the supernatant was taken for protein quantification with the Pierce™ BCA Protein Assay Kit (Thermo Scientific).

For the generation of tryptic peptides, we applied filter-aided sample preparation[47]. To be able to absolutely quantify the proteins in the samples we used a standard range of proteins (UPS2-1SET, Sigma), which we spiked into one of the samples (3.3 μg in sample equivalent to 100 000 cells)[48]. To obtain deep-proteomes, samples were fractionated using strong anion exchange, collecting fractions of the flow through and elutions at pH 11, 8, 5 and 2 of Britton& Robinson buffer. Samples were desalted and concentrated using C18 stage tips.

The peptide samples were separated on an Easy nLC 1000 (Thermo Scientific) connected online to a Thermo scientific Orbitrap Fusion Tribrid mass spectrometer. A 240 min acetonitrile gradient (5–23%, 8–27%, 9–30%, 11–32% and 14–32% for FT, pH 11, 8, 5 and 2, respectively) was applied to the five fractions. MS and MS/MS spectra were recorded in a Top speed modus with a run cycle of 3 s. MS/MS spectra were recorded in the Ion trap using higher-energy collision dissociation fragmentation. To analyse the raw MS data we used MaxQuant (version 1.5.1.0, database: Uniprot_201512/HUMAN)[49] with default setting and the match between runs and iBAQ algorithms enabled. We filtered out reverse hits and imputed missing values using Perseus (default settings, MaxQuant software package).

**CEL-seq2 mRNA quantification.** mRNA sequencing was performed according to the CEL-seq2 protocol[50] with adaptations. Reverse transcription was performed in 2 μl reactions overlaid with 7 μl Vapor-Lock (Qiagen) using Maxima H minus reverse transcriptase (Thermo Fisher) and 100 pg purified RNA per sample. Primer sequences were adapted to allow sequencing of 63 nucleotides of mRNA in read 1 and 14 nucleotides in read 2, comprising the sample barcode and UMI. Reverse transcription primer: 5′GCCGGTAATACGACTCACTATAGGGGTTCAGACGTGTGCTCTTCCGATCTNNNNNNNN[6ntsample-barcode]TTTTTTTTTTTTTTTTTTTTTTTTTTV3′, random-octamer-primer for reverse transcription of amplified RNA: 5′CACGACGCTCTTCCGATCTNNNNNNNNN3′, libraryPCRPrimers: 5′AATGATACGGCGACCACCGAGATCTACACTCTTTCCCTACACGACGCTTCCGATCT3′ and 5′CAAGCAGAAGACGGCATACGAGAT[6ntindex]GTGACTGGAGTTCAGACGTGTG-CTCTTCCGATC3′.

Sequencing was performed using the NextSeq500 from Illumina.

**Colony formation assay.** In six-well plate, 200 000 feeder (J2-3T3) cells were seeded in Dulbecco's modified Eagle's medium (DMEM; with 10% bovine serum (BS) and 1% pen/strep). After 1 day, feeder cells were inactivated by 3-h treatment with mitomycin C. After thorough washes with DMEM, 1000 keratinocytes were seeded into each well. The following day, treatment was started (day 0) by refreshing medium and addition of the indicated concentration of compound, or DMSO as a vehicle control. Cells were grown in the presence of compounds for 8 more days, and the medium was refreshed on days 2 and 5. Rocki was present until day 2 of the treatment. Cells were fixed, stained with TGM1-specific antibodies and scanned as described before[27]. Raw images from the LiCor Odessey system were processed with CC Photoshop and CellProfiler with consistent settings. Data obtained via automatic counting and imaging analysis via CellProfiler were analysed and visualised in the R programming language.

**Code availability.** R-package IDseq (version 0.1.0) is available from https://github.com/jessievb/IDseq. Colony-forming assay analysis scripts (Cell-profiler and R-script) are available from https://github.com/jessievb/automated_CFA.

**Data availability.** Used antibodies and oligo sequences are available as Supplementary Data 1 and Supplementary Data 3, respectively. Sequencing data and processed data from ID-seq experiments are available through GEO[43] Series accession number GSE100135. CEL-seq2 data to identify highly expressed kinases (Supplementary Fig. 19) are available upon reasonable request. Images from Supplementary Fig. 22 were obtained from Human Protein Atlas (Version 16.1).

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

## Acknowledgements

We would like to thank E. Janssen-Megens and S. van Genesen for technical support and discussions, Ana Pombo for discussions and critical reading of the manuscript. We would like to thank Bill Zuercher for discussions and reagents. The PKIS was supplied by GlaxoSmithKline, LLC and the Structural Genomics Consortium under an open access Material Transfer and Trust Agreement: http://www.sgc-unc.org. K.W.M., J.A.G.v.B, J.P.G., M.H. and S.E.J.T. are financially supported by the Radboud University, the Dutch Organisation for Scientific Research (NWO-VIDI) and the European Union (Marie-Curie Career Integration Grant). J.A.G.v.B was supported by the ERC grant ERC-2013-AdG No. 339431—SysStemCell (to H.G.S.)

## Author contributions

J.A.G.v.B. designed the experiments, designed ID-seq barcodes, performed the ID-seq experiments and colony formation assays, wrote the code and R-package and analysed the data. J.P.G. and M.H. performed the iPCR and RNA-seq experiments. S.E.J.T., P.W.T. C.J. and M.V. performed the proteomics analysis and provided reagents. S.E.J.T. performed nucleofections for siRNA-mediated knockdown. J.M. and R.v.d.S. produced all oligo DNA nucleotides. H.G.S. designed and analysed experiments. C.A.A. developed the model for ID-seq count data analysis. K.W.M. conceived and oversaw the study, designed experiments and analysed the data; K.W.M. and J.A.G.v.B. wrote the manuscript with input from all other authors.

## Additional information

**Competing interests:** J.M. and R.v.d.S. are employees of Biolegio BV, an SME producing and selling oligo DNA nucleotides. The remaining authors declare no competing interests.

