## [Peer Review File · Nature Communications]

Reviewers' comments:

Reviewer #1 (Remarks to the Author):

This is a very interesting paper describing a new screening technology based on oligo-tagged antibodies. A library of antibodies in which each antibody is labeled with a unique tag sequence is generated and used to identify epitopes across many samples in a high throughput format. The approach has the combined benefits of being highly multiplexable, allowing tens to hundreds of antibodies to be used simultaneously, and high throughput, since the separate samples are barcoded prior to sequencing, allowing all sample material to be pooled in a single run. The reagents also have other benefits for sensitivity and specificity, and incorporate UMIs that allow accurate quantitation, which is important for measuring levels of the different target epitopes.

The paper is well written and the figures are clear, organized and attractive. Thus, in addition to high quality production, I also see that this method could be important for a variety of screening applications, as has been demonstrated by examples in the manuscript.

I recommend publication without reservation.

Reviewer #2 (Remarks to the Author):

The manuscript by van Buggenum et al describes a new high-throughput cell culture based phenotype screening technology called Immuno-Detection by sequencing (ID-seq) that combines immune-staining and massive parallel sequencing. The authors develop both the experimental technology and computational pipeline allowing simultaneous analysis of effect of hundreds and even thousands compounds on the cellular phenotypes. The paper further explores the usage of the technology to validate the known and uncover new kinase-mediated pathways controlling epidermal progenitor cell maintenance and differentiation. The quality of data is very high, the figures and tables are well presented. The manuscript is of high interest for a broad readership of Nature Communications journal.

My only concern is rather very brief discussion of the results. I suggest the authors to extend discussion of the future usage of this exciting technology for different application beyond analysis of the kinase-mediated pathways, how it fits among the existent high throughput cell-based screening technologies and what a significance of the new insight into the epidermal keratinocyte biology revealed in the manuscript.

I recommend the manuscript for a publication in Nature Communications after the revision of the discussion section.

Reviewer #3 (Remarks to the Author):

The manuscript by van Buggenum and co-workers describes a new technology (called ID-seq) that allows multiplexing the immunodetection of various (phospho-)protein targets as biomarkers in fixed cells by combining antibody-DNA conjugates with high-throughput sequencing. Conceptually, the idea of generating antibody-DNA conjugates to multiplex readouts has been demonstrated by others as well as by the authors in previous publications. However, coupling it with high-throughput sequencing is where the novelty resides. This should provide a powerful means for rapidly and comprehensively screening the effect of a diverse set of conditions on biological samples.

The authors used this approach to document the effect of a panel of ~300 chemical compounds on

epidermal stem cell renewal and differentiation by measuring in parallel the fluctuations of 70 (phospho-)protein biomarkers. Their work suggested the implication of the mTOR pathway in epidermal differentiation. It also revealed the putative implication of a number of kinases in epidermal stem cell renewal.

This is definitely an exciting study as it offers great potential for enabling high-resolution “molecular phenotyping” of cells in a variety of conditions and sample formats. My enthusiasm however is somewhat dampened by a number of technical and conceptual issues that are listed below.

Major comments:

1- The authors do not sufficiently detail how the signal to noise ratio is determined. What is the sensitivity of their method? What is the dynamic range? How do these parameters compare with other quantitative detection methods such as AlphaScreen, Luminex, etc? For instance, in Fig. S5, the authors show the response of several of their readouts to an EGFR inhibitor or to EGF stimulation in skin stem cells. The response of pAKT-T308, pAKT1/2/3 and pERK to EGF stimulation (as estimated by the plots) are at best modest. One would have expected strong stimulation by EGF. Similarly, EGFR inhibition should result in complete inhibition of the target phosphorylation site (pY1068; Fig. S5). Yet, only a modest effect is detected.

Another instance is the TGM1 expression change upon EGFR inhibition (~1.4X fold-change; Fig. 2c). How does this modest change compare with published work? Was this corroborated by another means? It has been reported that TGM1 mRNA expression in differentiating human keratinocytes can go up by 80-90-fold. Careful benchmarking should be performed.

Finally, in parallel to using chemical inhibitors, the authors should benchmark their methodology (e.g., s/n ratio, dynamic range and sensitivity) using siRNA or (even better) CRISPR knockout on a limited set of targets. Considering the often questionable high-selectivity of a majority of antibodies, such controls would greatly help not only to assess the robustness of the method, it would also unequivocally assess signal specificity.

2- The contribution of non-specific signal due to hybridization of DNA barcodes with cellular nucleic acids is minimized by adding a single-stranded ssDNA blocking step. However, it is not clear how the possible non-specific capture of antibodies by DNA barcodes has been addressed.

3- On page 5, the authors mention a coefficient of variation of 20% (Fig. 1d). This appears to be high but consistent with the data shown in Fig. S5. Given a 20% CV, what would be the expected false-discovery rate (both false positive and false negative) associated to this methodology for modifications induced at a low stoichiometry?

4- It is not clear how the panel of biomarkers was selected. Apart from being constituted of validated antibodies, this is not clarified in the text. It appears to be tailored to detect markers of keratinocyte differentiation. If so, antibodies against keratin isoforms or filaggrin would have been relevant.

5- It is problematic that phospho-specific antibodies are not systematically accompanied by counterpart antibodies that recognize the non phosphorylated form in order to validate the modulated phosphorylation events. For instance, c-Fos, c-Jun, EGFR, MAPK p38 are only detected by a phospho-specific antibody. This is a major issue when it comes to normalizing the phosphorylation signal to changes in protein abundance. Especially in a context where stimulations are applied chronically for several days to study differentiation and where significant changes in protein abundance can occur.

6- The authors claim the discovery of the mTOR pathway as being involved in keratinocyte differentiation. However, this conclusion has not been validated by any alternative means, such as

standard western blot monitoring phospho and total signals for mTOR, Akt, 4E-BP, S6K, etc. Furthermore, the effect of the mTOR pathway should be confirmed by functional approaches other than chemical inhibitors (e.g. siRNA, CRISPR, etc.).

7- Data analysis by principal component aggregation and the described generalised linear mixed (glm) model are non-intuitive and confusing. Why not using more straightforward measures like fold-change and clustering to describe the results?

8- Validation of the newly identified kinases involved in keratinocytes differentiation is not convincing. Firstly, of the 10 dose-response curves presented in Fig. 6, only the response to the EGFR inhibitor AG1478 fits a classical sigmoidal function. The other inhibitors show very steep responses at the highest dose indicative of either multiple targets or non-specific effects on cell viability. Furthermore, the inhibitors chosen to validate specific kinases have unclear selectivities. For instance, the RSK inhibitor BI-D1870 was reported to inhibit several other targets including slk, lok and mst-1 (Edgar et al. 2013). These experiments should include more than one pharmacological agent per targeted kinase. Importantly, they should also functionally validate them using siRNA or CRISPR KO. Finally, the authors should also confirm target engagement by monitoring downstream phosphosite regulated by these kinases.

9- For PKIS data analysis, have the authors detected the effect of the kinase inhibitor probes on their bona fide targets? For instance, did the set of JAK inhibitors result in modulation of STAT phospho-signal? Idem for MEK inhibitors with respect to pERK, etc. This would have strengthened the use of ID-seq to detect genuine signaling effects. This data would deserve to be presented and commented.

10- At the bottom of p. 4, the authors mentioned that "From these 111 antibodies, 84 showed robust signals in In-Cell-Western/IF and/or immune-PCR experiments using antibody dilutions and/or IgG control antibodies." Where is the data for this claim? This is important in order to assess the quality of the selected antibodies.

11- On p. 5 (line 4), the authors say: "In addition, signals of a subset of phospho-specific antibodies were decreased upon phosphatase treatment of fixed cell populations." Again, there is no reference for this claim. Where is the data?

12- In their EGFR inhibition experiments, the authors used the AG1478 inhibitor at 10 micromolar. This seems to be excessive given that its reported IC50 is in the low nanomolar (3-5nM) range. How can they be sure that they are not also measuring off-target effects?

Minor comments:

1- The word "phenotype" is frequently used in the wrong context. For instance, on page 7, "We confirmed that these phenotypes truly reflect keratinocyte differentiation". The phenotype here is at best a "molecular phenotype" or a readout, not the appearance of an individual cell or organism influenced by its genotype. In other words, a molecular readout does not systematically translate into a cellular response.

2- In Figure 6, PF-4708371 should read PF-4708671.

We would like to thank all three referees for their constructive feedback and criticism. We have addressed the points that were raised to the best of our ability. Please find a point-by-point response to each of the comments offered per reviewer.

Reviewer #1 (Remarks to the Author):

This is a very interesting paper describing a new screening technology based on oligo-tagged antibodies. A library of antibodies in which each antibody is labeled with a unique tag sequence is generated and used to identify epitopes across many samples in a high throughput format. The approach has the combined benefits of being highly multiplexable, allowing tens to hundreds of antibodies to be used simultaneously, and high throughput, since the separate samples are barcoded prior to sequencing, allowing all sample material to be pooled in a single run. The reagents also have other benefits for sensitivity and specificity, and incorporate UMIs that allow accurate quantitation, which is important for measuring levels of the different target epitopes.

The paper is well written and the figures are clear, organized and attractive. Thus, in addition to high quality production, I also see that this method could be important for a variety of screening applications, as has been demonstrated by examples in the manuscript.

I recommend publication without reservation.

We thank reviewer #1 for the compliment on the quality of the clarity and organization of the manuscript and agree with the notion that ID-seq can play an important role for wide variety of screening applications like drug or other perturbations screens.

Reviewer #2 (Remarks to the Author):

The manuscript by van Buggenum et al describes a new high-throughput cell culture based phenotype screening technology called Immuno-Detection by sequencing (ID-seq) that combines immune-staining and massive parallel sequencing. The authors develop both the experimental technology and computational pipeline allowing simultaneous analysis of effect of hundreds and even thousands compounds on the cellular phenotypes. The paper further explores the usage of the technology to validate the known and uncover new kinase-mediated pathways controlling epidermal progenitor cell maintenance and differentiation. The quality of data is very high, the figures and tables are well presented. The manuscript is of high interest for a broad readership of Nature Communications journal.

My only concern is rather very brief discussion of the results. I suggest the authors to extend discussion of the future usage of this exciting technology for different application beyond analysis of the kinase-mediated pathways, how it fits among the existent high throughput cell-based screening technologies and what a significance of the new insight into the epidermal keratinocyte biology revealed in the manuscript.

I recommend the manuscript for a publication in Nature Communications after the revision of the discussion section.

We share the excitement in the potential applications of the ID-seq technology and have revised the discussion section considering the points raised by reviewer #2.

Reviewer #3 (Remarks to the Author):

The manuscript by van Buggenum and co-workers describes a new technology (called ID-seq) that

allows multiplexing the immunodetection of various (phospho-)protein targets as biomarkers in fixed cells by combining antibody-DNA conjugates with high-throughput sequencing. Conceptually, the idea of generating antibody-DNA conjugates to multiplex readouts has been demonstrated by others as well as by the authors in previous publications. However, coupling it with high-throughput sequencing is where the novelty resides. This should provide a powerful means for rapidly and comprehensively screening the effect of a diverse set of conditions on biological samples.

The authors used this approach to document the effect of a panel of ~300 chemical compounds on epidermal stem cell renewal and differentiation by measuring in parallel the fluctuations of 70 (phospho-)protein biomarkers. Their work suggested the implication of the mTOR pathway in epidermal differentiation. It also revealed the putative implication of a number of kinases in epidermal stem cell renewal.

This is definitely an exciting study as it offers great potential for enabling high-resolution "molecular phenotyping" of cells in a variety of conditions and sample formats. My enthusiasm however is somewhat dampened by a number of technical and conceptual issues that are listed below.

We are happy to read reviewers #3 excitement of the study, and the notion that ID-seq offers great potential in a wide variety of conditions and sample formats. Several of the issues raised are indeed important considerations. In a revised manuscript we will therefore provide several additional figures to address the technical issues raised, and we will modify the manuscript appropriately.

Below we provide a point-by-point response to the issues raised by this reviewer and indicate how these have been incorporated in the revised manuscript.

Major comments:

1- The authors do not sufficiently detail how the signal to noise ratio is determined. What is the sensitivity of their method? What is the dynamic range?

We thank the referee for pointing out the lack of detail in the description on technical properties like signal-to-noise, sensitivity and dynamic range of ID-seq. Therefore, we revised the manuscript to provide this information in the materials and methods and in the figure legends of Supplementary Figure S3 (dynamic range) and Supplementary Figure S10 (signal-over-background). For this latter point, also see the response to comment 2.

In addition to these textual revisions, we performed ID-seq experiments addressing sensitivity of the method and included this in the manuscript. We used a selected number of antibodies for ID-seq of a cell-dilutions series (Figure Revisions -1.1). The signals of the antibodies correlated with the decreased epitope presence down to only ~1250 cells per sample (of which only a fraction was used to prepare the sequencing sample). This illustrates that ID-seq detects a range of epitope abundances from extracellular and intracellular proteins. Due to limitations of cell seeding in the plate-setup we are not able to test lower cell numbers, though we do expect based on the mentioned dynamic range of the sequencing readout, that the sequencing readout of antibody-DNA conjugates is able to measure even lower epitope abundances. The results from these experiments have included in the revised manuscript as Figure S5a.

Figure Revisions -1.1 ID-seq signal from 7 antibodies from different number of keratinocytes. Cells were seeded at different cell numbers as indicated, and grown for 1 day before fixation and ID-seq analysis. UMI-counts show the expected 2-fold decrease based on seeded cell numbers.

How do these parameters compare with other quantitative detection methods such as AlphaScreen, Luminex, etc?

We agree that a comparison to other approaches is warranted and thank the reviewer for this suggestion. The dynamic range (4 orders of magnitude, Figure S3) of our counting by sequencing methods is comparable to the advertised range for the AlphaScreen assay (PerkinElmer) and the Luminex assay (RND systems). We note that there are some fundamental differences in these assays compared to ID-seq. First, AlphaScreen requires pairs of antibodies for each target, whereas ID-seq uses a single antibody per target. Second, both the AlphaScreen and Luminex assays are designed to be used on proteins in solution (such as bodyfluids and cell extracts). In contrast, ID-seq does not require cell lysis and works on fixed cells in multi-well plates. Therefore, we view ID-seq as a new technology that is very much complementary to the AlphaScreen and Luminex approaches. This point is now explicitly included in the discussion section of the revised manuscript.

For instance, in Fig. S5, the authors show the response of several of their readouts to an EGFR inhibitor or to EGF stimulation in skin stem cells. The response of pAKT-T308, pAKT1/2/3 and pERK to EGF stimulation (as estimated by the plots) are at best modest. One would have expected strong stimulation by EGF. Similarly, EGFR inhibition should result in complete inhibition of the target phosphorylation site (pY1068; Fig. S5). Yet, only a modest effect is detected.

We agree with reviewer 3 that the mentioned antibodies show modest effects. However, we respectfully disagree with the notion to expect strong stimulation by EGF because of the following reason. In our culture system of primary human keratinocytes, there is considerable autocrine EGF signalling activity. Therefore, we did not have a clear expectation of the effect of stimulation with supplemented EGF presented in Supplementary figure S5 (now figure S8). Unfortunately, we do not have an explanation for the lack of effect on pY1068 EGFR antibody by EGFR inhibition at this point.

Another instance is the TGM1 expression change upon EGFR inhibition (~1.4X fold-change; Fig. 2c). How does this modest change compare with published work? Was this corroborated by another means? It has been reported that TGM1 mRNA expression in differentiating human keratinocytes can go up by 80-90-fold. Careful benchmarking should be performed.

We agree with the reviewer that mentioned differentiation marker TGM1 can go up to 80-90 fold in some differentiation systems. It is important to note that TGM1 is a late differentiation marker, and the presented (protein) measurement is after only 48 hours of EGFR inhibition. At mRNA level we observed (not presented data) that TGM1 levels strongly increased after 72 and 96 hours of AG1478 treatment, suggesting 48 hours of AG1478 treatment does not yet reflect a fully differentiated keratinocyte.

We performed extra experiments (and included the data in revised manuscript) to verify that multiple independent read-outs reflect the observed increase in TGM1 levels at 48 hours of AG1478 treatment. Figure Revisions -1.2 shows TGM1 levels measured by direct epitope measurement via mass

spectrometry (proteomics dataset), immuno-fluorescence ('In Cell Western'), and mRNA levels via quantitative RT-PCR. We observe that ICW and qPCR show higher fold increase (~2,5-4 fold) than ID-seq and proteomics dataset. We note that these two experiments are corrected for potential differences in confluency and cell number. Slight differences in confluency and with this differentiation 'status' could cause differences in TGM1 levels of these experiments. The proteomics dataset shows consistent fold increase (~1.4x) like the ID-seq measurement. Therefore, we conclude that the observed modest change (upon 48 hours AG1478 treatment) reflects the protein level change and that these differences can be accurately identified using ID-seq. These results are included in the revised manuscript as Figure S12.

Figure Revisions -1.2 TGM1 protein or mRNA levels measured by multiple methods in AG1478 and vehicle control treated keratinocytes. **a.** ID-seq signal (n=3) and quantitative proteomics signal of TGM1 peptides (n=3) shows comparable dynamics. **b.** In cell western (immuno-fluorescent staining of fixed and permeabilized cell populations similar conditions as ID-seq) signal for TGM1 antibody corrected for DNA content (as proxy for cell numbers) shows larger fold increase in TGM1 than with proteomics or ID-seq. **c.** mRNA levels (corrected for 18S levels) of TGM1 show comparable dynamics as in cell western determining protein levels.

Finally, in parallel to using chemical inhibitors, the authors should benchmark their methodology (e.g., s/n ratio, dynamic range and sensitivity) using siRNA or (even better) CRISPR knockout on a limited set of targets. Considering the often questionable high-selectivity of a majority of antibodies, such controls would greatly help not only to assess the robustness of the method, it would also unequivocally assess signal specificity.

We agree that antibodies should be specific for the intended target, and therefore added extra benchmarking experiments to the revised manuscript. Although CRISPR knockout would be a great asset, these experiments are technically very challenging in primary epidermal stem cells. In our experience this requires selection single-cell derived clones, and often two rounds of gRNA targeting. In addition, knock-down of stem-cell related proteins can lead to cellular differentiation, and therefore a block in proliferation, we anticipate that obtaining CRISPR/Cas based knock-out cells will not be possible. We were therefore not confident that such an exercise would yield conclusive results. Instead, we performed siRNA-mediated knockdowns of five selected proteins that were included in our ID-seq antibody panel Figure revisions -1.4 shows qPCR and ID-seq signal of control and knockdown cell populations. The decreased ID-seq signals with all 5 antibodies after siRNA transfection matches the extent of mRNA knock-down. These results confirmed the specificity of the antibody-DNA conjugates used in ID-seq. This data is now included in the revised manuscript as Figure S5b.

Figure revisions -1.4. siRNA mediated knockdown of proteins results in reduced mRNA (qPCR) and protein (ID-seq) levels. (n=3)

2- The contribution of non-specific signal due to hybridization of DNA barcodes with cellular nucleic acids is minimized by adding a single-stranded ssDNA blocking step. However, it is not clear how the possible non-specific capture of antibodies by DNA barcodes has been addressed.

Indeed, minimizing non-specific signal is a very important point raised by reviewer #3. We have described and solved this problem in our previous publication (van Buggenum 2016, of which the relevant Figure 7 is presented below). To address the point of background staining, we used immuno-PCR experiments to determine the signal over background (as defined comparing the antibody-DNA signal to both an empty well (no-cells), as well as control IgG antibody conjugates) using a range of blocking conditions. Based on these experiments we found that the addition of (single-stranded) salmon sperm DNA and protein-free blocking buffer greatly reduces non-specific DNA-barcode signals (likely by obstructing non-specific capture sites). Importantly, this is reflected by the fact that the control IgG signals were the same as empty wells. For most of the antibodies used in our ID-seq panel, this signal over noise ratio ranges from 5-500 fold (presented Figure S11 in the revised manuscript). We feel confident that non-specific capture of the antibody-DNA barcodes is minimized as much as possible. We have now described how we determined the signal to noise ratio more clearly in the manuscript.

Fig 7, van Buggenum et. al. (Scientific Reports 2016)

3- On page 5, the authors mention a coefficient of variation of 20% (Fig. 1d). This appears to be high but consistent with the data shown in Fig. S5. Given a 20% CV, what would be the expected false-

discovery rate (both false positive and false negative) associated to this methodology for modifications induced at a low stoichiometry?

We respectfully disagree that a CV of 20% is high. A coefficient of 20% we generally regarded as an indication of a very robust assay. Using 20 simulations of samples with a CV of 20% (n=100), we estimate the FNR (% of non-significant statistical tests between a distribution of an average of 100 counts versus the other averages, 5 count step-size) to be ~5%. In contrast, the false positive rate was determined as the % of positive statistical tests among 100 simulations of random distributions with the same average value and a CV of 20%. This amounted to a median FPR of ~ 1%. This indicates that our ID-seq technology can robustly and sensitively detect even small differences among samples.

4- It is not clear how the panel of biomarkers was selected. Apart from being constituted of validated antibodies, this is not clarified in the text. It appears to be tailored to detect markers of keratinocyte differentiation. If so, antibodies against keratin isoforms or fillagrin would have been relevant. *We thank the reviewer for pointing this out and apologise for this omission. We revise the text to clarify how the panel of antibodies was selected. To clarify, we selected a panel of antibodies that can be used in a wide variety of systems, including several signalling pathways and cell-cycle marker for example. We took care to include mostly antibodies that are proven or very likely to detect both the human as well as the mouse epitopes. Therefore, our panel of antibodies should serve as a useful resource and starting point for researchers working in other cell systems as well. As we work with keratinocytes, we did add selection of antibodies targeting keratinocyte specific proteins, which allowed us to detect changes in two integrin levels, and one differentiation protein TGM1. Our antibody selection can be viewed as a proof-of-principal panel that can be modified and supplemented to meet the needs of individual projects.*

5- It is problematic that phospho-specific antibodies are not systematically accompanied by counterpart antibodies that recognize the non phosphorylated form in order to validate the modulated phosphorylation events. For instance, c-Fos, c-Jun, EGFR, MAPK p38 are only detected by a phospho-specific antibody. This is a major issue when it comes to normalizing the phosphorylation signal to changes in protein abundance. Especially in a context where stimulations are applied chronically for several days to study differentiation and where significant changes in protein abundance can occur.

We agree with the reviewer that our panel indeed does not include the non-phosphorylated versions for all phospho-specific antibodies. For most we did try to include these, but were unable to verify their selectivity to a high enough confidence and therefore choose not to include them in our current panel of antibodies. However, our current panel does include counterparts for smad1, smad2, smad3, smad5, FAK, JNK, c-Myc, Cdc2, ERK12, Akt2, Ephrin B3, however not for p-S6K for example. We agree that several days treatment can induce protein abundance change and correction of phospho-protein levels for the total levels of that protein will prove to be important for certain applications. For these applications the individual researchers will need to supplement their antibody panel with the appropriate control antibodies, as will we in future projects.

6- The authors claim the discovery of the mTOR pathway as being involved in keratinocyte differentiation. However, this conclusion has not been validated by any alternative means, such as standard western blot monitoring phospho and total signals for mTOR, Akt, 4E-BP, S6K, etc. Furthermore, the effect of the mTOR pathway should be confirmed by functional approaches other than chemical inhibitors (e.g. siRNA, CRISPR, etc.).

We thank the reviewer for the suggestion to further investigate this observation. For these experiments we choose to employ an epidermal differentiation method that is independent from chemical inhibition of signaling pathways, to prevent confounding effects. Instead, we used confluency to activate the differentiation process. Using quantitative immunofluorescence (In-Cell-Western) we confirmed induction of the late differentiation marker TGM1 with increasing cell density (Figure revisions 6.1a). Moreover, PPL and TGM1 mRNA levels were similarly increased, further

confirming differentiation in these samples (Figure revisions 6.1b). Conversely, we observed a confluency dependent decline of phosphorylated-S6 levels (quantitative immunofluorescence) that is fully consistent with our original observation that decreased mTOR signaling is associated with epidermal differentiation (Figure revisions 6.1a). To gain more insight into the potential mechanism underlying this decreased activity we investigated the expression levels of mTOR and its co-factor RAPTOR. Strikingly, RAPTOR expression was inversely correlated with confluency induced differentiation, suggesting that regulation of RAPTOR expression during differentiation potentially, partially, explains our observations. However, it seems that decreased RAPTOR expression on its own is not sufficient to induce terminal differentiation in human keratinocytes as siRNA mediated silencing of RAPTOR did not affect mRNA levels of the differentiation markers IVL, PPL and TGM1 (Figure revisions 6.2). Consistent with our findings, mTOR signalling pathway was recently described to play a role in the switch from proliferation to differentiation in human keratinocytes (Buerger et.al., Plos One, 2017). We revised the manuscript text (new data is included as Figure S8) and discussion to include these additional experiments of the role and regulation of the mTOR pathway during epidermal differentiation. These additions serve as a good illustration of how the application of ID-seq can lead to new insights into biological processes.

Figure revisions 6.1. Confluency induced differentiation of keratinocytes shows increased differentiation and decreased mTOR signalling activity. (a) immuno-fluorescence (in-cell-western) signal corrected for cell number via DRAD5 staining of transglutaminase 1 and phospho-S6 kinase. (b) RT-qPCR shows mRNA levels (corrected for 18S) of differentiation markers periplakin (PPL) and transglutaminase 1 (TGM1) and mTOR signalling components mTOR and RAPTOR.

Figure revisions 6.2. mRNA levels show that siRNA knockdown of RAPTOR does not result in increased expression levels of involucrin (IVL) periplakin (PPL) or transglutaminase 1 (TGM1) (n=3, after 4 days of nucleofection).

7- Data analysis by principal component aggregation and the described generalised linear mixed (glm) model are non-intuitive and confusing. Why not using more straightforward measures like fold-change and clustering to describe the results?

There are several reasons why we choose the glm model, and PCA analysis. First, the glm model allows us to use appropriate statistical testing considering negative binomial distribution of the UMI data while regressing-out differences in sequencing depth, with potential plate or batch effects. Second, the PCA analysis includes multiple antibodies into one measure, 'summarizing' the biological state based on multiple measurements which cannot be achieved by assessment of antibodies one-by-one. Thus, a more straight-forward approach would not yield the same depth of insights as the approach we took.

8- Validation of the newly identified kinases involved in keratinocytes differentiation is not convincing. Firstly, of the 10 dose-response curves presented in Fig. 6, only the response to the EGFR inhibitor AG1478 fits a classical sigmoidal function. The other inhibitors show very steep responses at the highest dose indicative of either multiple targets or non-specific effects on cell viability. Furthermore, the inhibitors chosen to validate specific kinases have unclear selectivities. For instance, the RSK inhibitor BI-D1870 was reported to inhibit several other targets including slk, lok and mst-1 (Edgar et al. 2013). These experiments should include more than one pharmacological agent per targeted kinase. Importantly, they should also functionally validate them using siRNA or CRISPR KO. Finally, the authors should also confirm target engagement by monitoring downstream phosphosite regulated by these kinases.

We fully agree it is of importance to keep in mind compounds may have effects on cell viability and/or may not be 100% selective for the intended target(s). We would like to note that cell viability in the presented colony formation assays may not be a major concern for the presented concentrations of compounds (up to 4 μM) as in our colony formation assay the number of colonies is not affected for 4/5 compounds (Figure revisions 8.1). Indeed at 20 μM 4 out of 5 compounds did affect cell viability and therefore these concentrations were excluded from the analysis. We have included these results as an additional supplemental figure to the revised manuscript (Figure S23).

Figure revisions 8.1 **a**. DRAQ5 (DNA staining) images of colonies treated with indicated concentration of indicated compounds. **b**. Quantification of number of colonies (from $n=3$ replicates).

Furthermore, we agree with the reviewer that chemical inhibitors may have affinity for other targets. We cannot exclude that the observed effects for mentioned RSK1/2/3/4 inhibitor (which indeed has multiple intended targets RSK1/RSK2/RSK3/RSK4) is influenced by inhibition of other mentioned kinases. The potential other targets that were indicated by the reviewer indeed seem to be expressed in keratinocytes (Figure revisions 8.2) and may thus potentially contribute to the observed effects, although these kinases were not identified by our statistical analyses.

Figure revisions 8.2 mRNA sequencing genome tracks from genes SLK, Lok and mst-1, from 0, 3 and 6 days contact induced differentiated keratinocytes.

We note that the suggested follow-up studies to test effect of specific knockdown or knockout are in principle different from inhibiting the kinases activity. As noted before, although CRISPR knockout would be an asset, these experiments are technically extremely challenging in primary epidermal stem cells. From our own practical experience this requires selection of single-cell derived clones, and often two rounds of gRNA targeting. Moreover, as deletion of the identified kinases will lead to cellular differentiation (and therefore a block in proliferation), we anticipate that obtaining CRISPR/Cas based knock-out cells will not be possible. We are therefore not confident that such an

exercise will likely yield conclusive results. Instead, we attempted siRNA-mediated knock-down of the identified kinases followed by colony formation assays or RT-qPCR. These experiments were performed 4-5 times, each time with adjustments to try and improve the quality. We have a lot of experience with siRNA-mediated knock-down in primary human epidermal keratinocytes (examples: Mulder NCB 2012, van Buggenum Sci Rep 2016, Figures on page 5 and 8 of this document) it seems that these targets are particularly refractory to knock-down. We obtained between 0 and 30 % knockdown of these kinases (in all independent attempts), while positive control showed >80 % knockdown. Unfortunately, these knockdown efficiencies are not sufficient to draw solid conclusions and therefore we regret we were not able to address this point of the reviewer within the 3 months of revision time.

9- For PKIS data analysis, have the authors detected the effect of the kinase inhibitor probes on their bona fide targets? For instance, did the set of JAK inhibitors result in modulation of STAT phospho-signal? Idem for MEK inhibitors with respect to pERK, etc. This would have strengthened the use of ID-seq to detect genuine signaling effects. This data would deserve to be presented and commented.

We agree with the reviewer that this data is present within the dataset and should be presented as supplementary data to the manuscript and modified the manuscript accordingly. In brief, the inhibitor library used for this screen is biochemically tested (by Elkins J. et.al. 2016). Based on this data we assigned (potential) inhibitors to kinases and in this way created 'sets of inhibitors' per kinase. Figure revisions 9 show for several of these inhibitor sets (EGFRi, RSK1i, RSK2i, p70S6Ki, p38ai, MAPKAPK3i, JNK2i) and the mean ID-seq signal of relevant downstream or upstream (phospho-)proteins. The compounds that can inhibit the EGFR clearly decrease phosphorylation levels of the EGFR, mTOR, cFOS and AKT (Figure revisions 9a). Interestingly, compound targeting RSK1/2 and/or p70S6K seem to increase phosphorylation of upstream EGFR and decrease S6K phosphorylation (Figure revisions 9b). Finally, phosphorylation of MKK3 is increased by compounds targeting MAPKAPK3, p38a and/or JNK2 (Figure revisions 9c). In all, these results confirm the effect of sets of PKIS compounds on relevant compounds.

Figure revisions -9. Effects of groups of inhibitors (x-axis) on indicated phospho-protein levels (panel title).

10- At the bottom of p. 4, the others mentioned that "From these 111 antibodies, 84 showed robust signals in In-Cell-Western/IF and/or immune-PCR experiments using antibody dilutions and/or IgG control antibodies." Where is the data for this claim? This is important in order to assess the quality of the selected antibodies.

We apologise for this omission and thank the reviewer for the suggestion to include this data. Supplementary table 1 (presented in the revised manuscript) summarises all the available data per antibody. Figure revisions 10.1 shows examples of In-Cell-Western antibody dilutions, and figure revisions 10.2 shows immuno-PCR from antibody dilutions series. This data is now included in the revised manuscript as Figures S6 and 7, respectively.

Figure revisions -10.1. Immuno-fluorescence staining of indicated (phospho-)proteins of primary human keratinocytes.

Figure revisions -10.2. Immuno-PCR signal of indicated antibodies (using different concentrations).

11- On p. 5 (line 4), the authors say: "In addition, signals of a subset of phospho-specific antibodies were decreased upon phosphatase treatment of fixed cell populations." Again, there is no reference for this claim. Where is the data?

We thank the reviewer for pointing this out. We have added a reference to supplementary table 1 with a list of all available data on each antibody. Figure revisions 11 shows reduced signal upon phosphatase treatment of several example antibodies, which is now included in the revised manuscript as Figure S9

Figure revisions -11 Immunofluorescent signal of indicated antibodies (panel captions) of fixed keratinocytes untreated or treated with Lambda phosphatase.

12- In their EGFR inhibition experiments, the authors used the AG1478 inhibitor at 10 micromolar. This seems to be excessive given that its reported IC50 is in the low nanomolar (3-5nM) range. How can they be sure that they are not also measuring off-target effects?

We respectfully disagree that the used concentration of AG1478 is excessive. We agree that it is important to use a concentration suitable for cell-based inhibition of the targeted molecule. The suggested IC-50 was tested biochemically, however the concentration we used is within the IC-50 range applied in multiple cell-based assays (Table revisions 1).

Table revisions 1. By selleckchem documented IC-50 in cell-based assays (<http://www.selleckchem.com/products/ag-1478-tyrphostin-ag-1478.html>)

Cell Lines	Assay Type	Concentration	Incubation Time	Formulation	Activity Description	PMID
U87MG	Growth inhibitory assay	~100 µM		DMSO	IC50=34.6 µM	8752145
U87MG ΔEGFR	Growth inhibitory assay	~100 µM		DMSO	IC50=8.7 µM	8752145
U87MG wtEGFR	Growth inhibitory assay	~100 µM		DMSO	IC50=48.4 µM	8752145
U87MG	Kinase assay	~100 µM		DMSO	inhibits EGFR tyrosine kinase activity	8752145
U87MG ΔEGFR	Kinase assay	~100 µM		DMSO	inhibits EGFR tyrosine kinase activity	8752145
U87MG wtEGFR	Kinase assay	~100 µM		DMSO	inhibits EGFR tyrosine kinase activity	8752145
HPV 16-immortalized human keratinocytes	Growth inhibitory assay	~50 µM		DMSO	inhibits cell growth	9288782
HPV 16-immortalized human keratinocytes	Function assay	~50 µM		DMSO	induces arrest in the Cell Cycle	9288782
HPV 16-immortalized human keratinocytes	Apoptosis assay	~50 µM		DMSO	induces apoptosis.	9288782
A431	Kinase assay	~10 µM		DMSO	inhibits the basal and TGF-α-stimulated tyrosine phosphorylation of the EGFR	10702262
MDA-468	Kinase assay	~10 µM		DMSO	inhibits the basal and TGF-α-stimulated tyrosine phosphorylation of the EGFR	10702262
A431	Function assay	~10 µM		DMSO	induces cell cycle arrest	10702262
MDA-MB-231	Kinase assay	~5 µM		DMSO	inhibits EGF stimulated phosphorylation of FKHR	11030146
CNE2	Growth inhibitory assay	100 µM		DMSO	inhibits cell proliferation by 98.4%	11410322
CNE2	Kinase assay	~100 µM		DMSO	inhibits EGFR tyrosine phosphorylation	11410322
CNE2	Function assay	~100 µM		DMSO	Inhibits MAPK and AKT activation	11410322
CNE2	Function assay	~50 µM		DMSO	affects cell cycle distribution	11410322
HSC-2	Kinase assay	8 µM		DMSO	inhibits phosphorylation of EGFR and Akt	17689285
HSC-2	Apoptosis assay	8 µM		DMSO	inhibits Fas-mediated apoptosis	17689285
HEp-2	Growth inhibitory assay	~10 µM		DMSO	enhances oridonin-induced growth-inhibitory	20202741
SubG1	Apoptosis assay	~10 µM		DMSO	enhances oridonin-induced apoptosis	20202741
HEp-2	Function assay	~10 µM		DMSO	enhances Oridonin-induced Bax activation, Bcl-2 degradation and SIRT1 inactivation	20202741
H508	Growth inhibitory assay	~1 µM		DMSO	mitigates CPF-mediated H508 cell growth	26514924

Minor comments:

1- The word "phenotype" is frequently used in the wrong context. For instance, on page 7, "We confirmed that these phenotypes truly reflect keratinocyte differentiation". The phenotype here is at best a "molecular phenotype" or a readout, not the appearance of an individual cell or organism influenced by its genotype. In other words, a molecular readout does not systematically translate into a cellular response.

We agree that this needs specification and revised the manuscript accordingly.

2- In Figure 6, PF-4708371 should read PF-4708671.

We thank the reviewer for noting this typo and revised the figure.

REVIEWERS' COMMENTS:

Reviewer #3 (Remarks to the Author):

The authors have satisfactorily answered all my comments.

Reply to reviewer's comments

REVIEWERS' COMMENTS:

Reviewer #3 (Remarks to the Author):

The authors have satisfactorily answered all my comments.

We thank reviewer #3 for useful comments to improve the manuscript and are happy to have answered all comments of reviewer 3.